# Spatiotemporally controlled genetic perturbation for efficient large-scale studies of cell non-autonomous effects

**Andrea Chai†, Ana M Mateus†, Fazal Oozeer, Rita Sousa-Nunes\***

Centre for Developmental Neurobiology, Institute of Psychiatry, Psychology and Neuroscience, King's College London, London, United Kingdom

**Abstract** Studies in genetic model organisms have revealed much about the development and pathology of complex tissues. Most have focused on cell-intrinsic gene functions and mechanisms. Much less is known about how transformed, or otherwise functionally disrupted, cells interact with healthy ones toward a favorable or pathological outcome. This is largely due to technical limitations. We developed new genetic tools in *Drosophila melanogaster* that permit efficient multiplexed gain- and loss-of-function genetic perturbations with separable spatial and temporal control. Importantly, our novel tool-set is independent of the commonly used GAL4/UAS system, freeing the latter for additional, non-autonomous, genetic manipulations; and is built into a single strain, allowing one-generation interrogation of non-autonomous effects. Altogether, our design opens up efficient genome-wide screens on any deleterious phenotype, once plasmid or genome engineering is used to place the desired miRNA(s) or ORF(s) into our genotype. Specifically, we developed tools to study extrinsic effects on neural tumor growth but the strategy presented has endless applications within and beyond neurobiology, and in other model organisms.

DOI: https://doi.org/10.7554/eLife.38393.001

**\*For correspondence:**
rita.sousa-nunes@kcl.ac.uk

†These authors contributed equally to this work

**Competing interests:** The authors declare that no competing interests exist.

## Introduction

Despite numerous and versatile genetic mosaic strategies available for genetically amenable model organism *Drosophila melanogaster*, none up to now was suited for efficient large-scale screening for cell non-autonomous effects on a developmentally deleterious genotype. Given the requirement for combinations of genetic manipulations, non-autonomous effects are more challenging to investigate yet well known to play crucial roles in development and disease contexts such as cancer. The challenge applies to any tissue but is particularly evident in the central nervous system (CNS) due to diversity of cell types and uniqueness of each lineage with respect to gene expression, size, projection patterns, as well as lethality frequently associated with their disruption. A much needed, transformative, new tool would be: (i) a viable parental stock in which (ii) chosen individual lineages could be (iii) triggered to assume a deleterious genotype (iv) with temporal control (v) from which point they would become permanently labeled by a reporter and (vi) with which a single cross to existing stocks would produce progeny with genetically perturbed cell types of interest other than the labeled lineages. To illustrate in our specific case: no available genetic tool allowed large-scale screening for non-autonomous effects on neural tumor growth as animals harbouring neural tumors cannot be kept as a stable stock.

*Drosophila* has been a canvas for pioneering mosaic tools, at the heart of which lie heterologous binary systems for transcriptional activation or recombination (*Griffin et al., 2014*). Transcriptional activation systems include the yeast transcription factor GAL4 and its binding site, named Upstream Activating Sequence (UAS); the bacterial LexA/LexA Operator (LexAop); and the fungal QF/QUAS system (*Brand and Perrimon, 1993*; *Yagi et al., 2010*; *Potter et al., 2010*). Recombination systems

include the bacteriophage Cre recombinase and its loxP target; the yeast Flippase/Flippase Recognition Target sites (FLP/FRT) and its variant mFLP5/mFRT71; and other yeast recombinases (KD, R, B2, and B3) and their cognate recognition sites (*Golic and Lindquist, 1989*; *Siegal and Hartl, 1996*; *Hadjieconomou et al., 2011*; *Nern et al., 2011*). The modularity of binary systems grants them combinatorial flexibility, and ingenious Boolean logic gates between recombination and transcriptional activation/silencing systems have expanded their applications (eg., *Struhl and Basler, 1993*; *Lee and Luo, 1999*; *Griffin et al., 2009*; *Yu et al., 2009*; *Yagi et al., 2010*; *Hadjieconomou et al., 2011*; *Hampel et al., 2011*); *von Philipsborn et al., 2011*; *Awasaki et al., 2014*). Binary systems have been extensively employed to perform large-scale screens using publically available UAS libraries to provide molecular understanding into numerous conserved cell intrinsic processes (*St Johnston, 2002*; *Kawakami et al., 2016*). Genome-wide screens remain to be applied to extrinsic processes modifying an adverse genotype.

We wished to determine the effects of microenvironment or systemic cues on tumor progression. To this end we needed to generate reproducible neural tumors in order to quantitatively assess growth. Tumor reproducibility requires control over lineage, induction time and consistency of levels of downregulation of tumor-suppressors and/or upregulation of oncogenes. We therefore aimed at generating tumors in restricted lineage subsets with a fast inducing event in parental (F0) animals, independently of GAL4/UAS so that we might employ this binary system (for which most modules exist in *Drosophila*, including for near genome-wide gain- and loss-of-function, readily available to the community) to cause non-autonomous perturbations on F1 progeny. Due to possible fate transformations and expression-level variations of regulatory sequences, we wanted tumors to become irreversibly labeled under the control of a ubiquitous and strong regulatory sequence from the time of induction. Various but not all of these features can be achieved with suppressible/inducible LexA, Q and FLP systems (*Weigmann and Cohen, 1999*; *Yagi et al., 2010*; *Riabinina et al., 2015*). Maintenance of an F0 stock with capacity for tumor induction requires suppression of the deleterious genotype until desired. However, whilst the lexA$^{GAD}$ derivative (superscript indicating the GAL4 activation domain) can be suppressed by GAL80, it is not compatible with continuous non-autonomous gene inductions via GAL4 as these would also be affected. Also, alleviation of QF suppression by quinic acid, or estrogen induction of FLP$^{EBD}$ (superscript indicating an estrogen-binding domain) requires ingestion and metabolization of the effector molecule, resulting in relatively long induction kinetics and variability, thus impairing reproducibility in the fast-developing fly tumor models (*Weigmann and Cohen, 1999*; *Potter et al., 2010*).

Our design presented achieves the desired features via the employment of two very efficient transcriptional termination sequences (STOP cassettes) upstream of an oncogenic sequence and reporter. Each STOP cassette is flanked by recombinase target sequences selective for two distinct recombinases, one constitutively expressed in selected lineages, conferring spatial specificity; the other whose expression is induced by heat-shock (hs), conferring rapid temporal resolution. We tested and refined the new genetic tools by recapitulating two well-established *Drosophila* neural tumor models, one generated by downregulation of the homeodomain transcription factor Prospero (Pros), which can lead to tumorigenesis in all neural lineages (of which there are around 100 per central brain lobe); another by downregulation of the NHL-domain protein Brain tumor (Brat), whose depletion leads to tumorigenesis specifically in so-called type II lineages (of which there are eight per brain lobe (*Figure 1—figure supplement 1*) (*Sousa-Nunes et al., 2010*). Starting from the units presented here our design can be multiplexed beyond two to produce further spatial intersections, or multiple temporal steps, along with any assemblies of gene expression downstream (downregulation and/or upregulation, plus reporter labeling). This strategy is therefore of broad interest, applicable to other tissues, organisms and biological questions, opening-up large-scale screening for non-autonomous effects.

## Results

### *FOFO* tool design features

Key to the design of this tumor-generating tool is that expression of deleterious sequences by the ubiquitous strong *actin5C* promoter, was blocked by not one (as commonly done), but two stringent STOP cassettes. Each STOP cassette was flanked by the selective recombination sites FRT and

mFRT71, specifically recognized by FLP and mFLP5, respectively (*Hadjieconomou et al., 2011*). We called this design 'FOFO', for Flp-Out-mFlp5-Out. The prediction was that expression would be unblocked only in the presence of the two Flippases, with spatiotemporal control achieved by lineage-restricted expression of FLP and hs-induction of mFLP5 (*Figure 1a*).

We wanted our tumor-generating tool to induce expression not only of oncogenes but to also allow downregulation of tumor suppressors, in addition to a reporter gene (in this case enhanced green fluorescent protein, EGFP). Multicistronic expression of oncogenic and reporter proteins can be easily achieved by sandwiching T2A peptide (*González et al., 2011*; *Diao and White, 2012*) codons between coding sequences (cds). We therefore focused on achieving a layout that reconciled strong reporter expression with gene downregulation by short hairpin artificial microRNAs (miRs). Artificial miRs consist of 21 bp sequences designed for RNA interference, embedded into a sequence backbone of a naturally occurring miR; they are very effective in downregulating gene expression (more so than long double-stranded RNAs; *Ni et al., 2011*), can be transcribed by RNA polymerase II (Pol II) (*Lee et al., 2004*), and can be concatenated for synergistic effect (*Chen et al., 2007*). We placed the EGFP cds downstream of an intron as this increases transcript expression (*Haley et al., 2010*) and has the additional advantage of being able to host miRs without disrupting transcript stability by their processing, unlike when miRs are placed in the 3' untranslated region (3'UTR) (*Bejarano et al., 2012*).

Wishing to study strictly cell non-autonomous effects employing the GAL4/UAS system, we included miRs targeting GAL4 as well as those targeting a tumor suppressor (two miRs per target). Therefore, if the GAL4 expression domain overlapped with the tumor domain, GAL4 would be silenced within the tumor. miRs targeting the neural tumor suppressors *pros* or *brat* were used for tumor induction and those targeting CD2 were used as control (*Yu et al., 2009*). To minimize position effects and enhance expression, all constructs generated for this study were flanked by gypsy insulators and integrated into the *Drosophila* genome by PhiC3-mediated transgenesis, selecting sites reported to produce low basal and high induced expression (*Markstein et al., 2008*).

The utility of this design lies in its combination with two distinct Flippases plus a desired GAL4 transgene in a single organism (*Figure 1b,F0* left). Once assembled, this stock can then be crossed to any other carrying a UAS-transgene (*Figure 1b,F0* right). The spatially restricted FLP will excise the first STOP cassette with a domain reproducibility that depends on enhancer reliability and strength as well as efficacy of the excision activity. In any case, neither reporter nor deleterious sequences should be expressed due to the additional STOP cassette. Consequently, until heat-shock, F0 and its F1 progeny should contain a single mFLP5-Out cassette within the FLP-expressing domain. F1 should also express the UAS-transgene in the GAL4 domain, and not express the miRs or reporter (*Figure 1b,F1* left). Following F1 heat-shock (*Figure 1b,F1* middle), the mFRT71-flanked STOP cassette should be excised (without spatial constraints, its efficacy depending on heat-shock duration); following which the miRs and reporter can be expressed but only within the FLP-expressing domain (*Figure 1b,F1* right). If the GAL4 domain overlaps with the FLP spatial domain (as schematized in *Figure 1b*), strictly non-autonomous effects can still be studied since GAL4 expression will be wiped-out therein by the GAL4$^{miRs}$ (*Figure 1b,F1* right). A more naturalistic schematic illustrating brain tumours and GAL4 driven in all glia is depicted in *Figure 1c*.

## Efficacy of STOP cassettes

Central to the success of this strategy is the efficacy of the STOP cassettes. For each, we used tandem transcriptional terminators, as others before us. Whereas some degree of STOP leakiness can be afforded to simply label cells or to generate a deleterious genetic perturbation by means of a cross, it is absolutely incompatible with our aim of harbouring a 'locked' deleterious perturbation within a stable stock. We tested a few transcriptional terminators until we obtained the tightly controlled expression necessary.

Removal of the *lamin* cds from the STOP cassette used in Flybow (*Hadjieconomou et al., 2011*) resulted in failure to terminate transcription despite concatenated *hsp70Aa* and *hsp27* terminators, seen by EGFP expression in the absence of Flippase (data not shown). In contrast, concatenation of *hsp70Bb* and *SV40* terminators, successfully precluded unintended EGFP expression. We therefore created a version of FOFO (FOFO1.0) with the two STOPs identical to the latter (*Figure 2a*). FOFO1.0 was tested with publicly available stocks of FLP and mFLP5 both under the control of the strong hs promoter. Encouragingly, in the presence of both hs-FLP and hs-mFLP5 and only after hs,

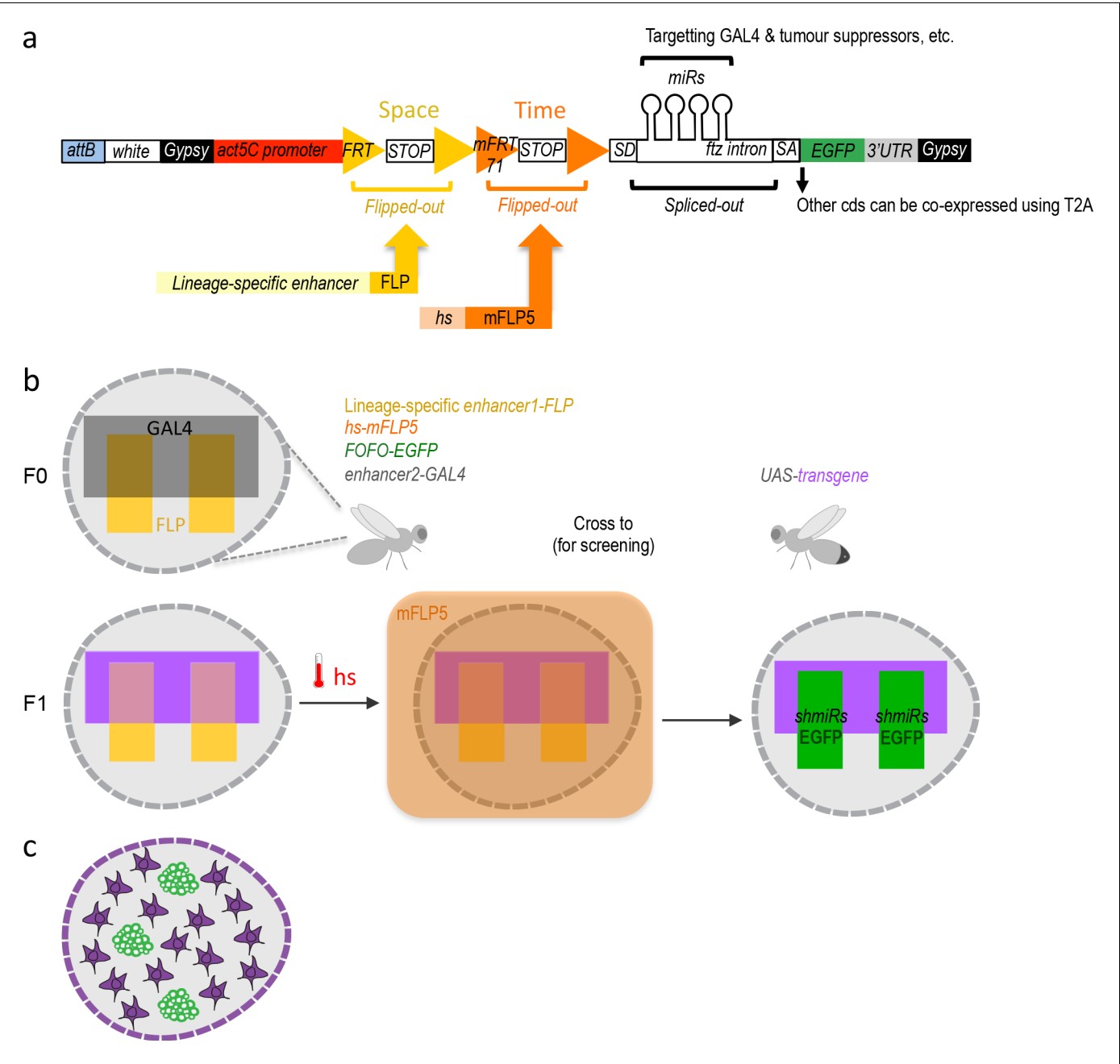

**Figure 1.** FOFO design and application. (**a**) FOFO construct design: the actin5C promoter is blocked from inducing transcript expression by two efficient transcriptional terminator (STOP) cassettes. Each of these is flanked by FRT or mFRT71, specifically recognized by FLP and mFLP5, respectively. Therefore, miRs and EGFP will only be expressed in cells containing the two flippases. Spatial and temporal control is achieved by providing a spatially restricted FLP and hs-induced mFLP5. SD, splice donor; SA, splice acceptor. Following excision of the *fushi tarazu* (*ftz*) intron, miRs are processed without detriment to reporter expression. Gypsy insulators minimize position effects whilst enhancing expression levels; attB sites allow site-specific insertion into attP-containing host strains. (**b**) Schematic of FOFO application. With the insertion sites chosen for this study, flies of the following genotype can be generated: *enhancer-FLP; hs-mFLP5, FOFO-EGFP; enhancer2-GAL4* (exemplifying with the GAL4 transgene on the third chromosome, though it could be placed elsewhere). Expression of deleterious sequences (either knock-down by miRs or overexpression alongside the reporter by means of T2A) can be induced (by heat-shock) in a single fly stock (without need to cross) carrying FOFO, a lineage-specific enhancer1-FLP and hs-mFLP5. The point is then to add in the same flies (F0 generation) a GAL4 transgene (enhancer2-GAL4) and cross to UAS responders. The FOFO containing stock expresses FLP in the spatially restricted domain defined by enhancer1 (yellow) in a tissue represented by the grey shape. FLP expression will constitutively excise the first STOP cassette but the presence of a second STOP cassette precludes expression of anything downstream unless flies are subject to hs. The F1 progeny expresses a transgene (purple) in the GAL4-expressing domain defined by enhancer2 (black). Following hs, mFLP5 expression leads to excision of the second STOP cassette and thus expression of miRs and EGFP in the domain covered by the lineage-

*Figure 1 continued on next page*

*Figure 1 continued*

specific enhancer. Even if the domain of the latter overlaps with that of enhancer2 as depicted, GAL4 miRs will delete GAL4 expression in the EGFP-expressing domain so that the GAL4 domain never overlaps with that of enhancer1 and only cell nonautonomous effects are assessed. (**c**) Schematic representation of a FOFO application with the tools designed for this study. EGFP-labeled neural tumors (green) are generated within brain lobes (grey shape) in a stock also carrying a GAL4 expressed in glia (purple). Crossing this stock to any UAS-responder lines (could be genome-wide gain- or loss-of-function) will allow identification of genes whose glial expression affects tumor size.

DOI: https://doi.org/10.7554/eLife.38393.002

The following figure supplement is available for figure 1:

**Figure supplement 1.** Schematics of *Drosophila* CNS and NSC lineages.

DOI: https://doi.org/10.7554/eLife.38393.003

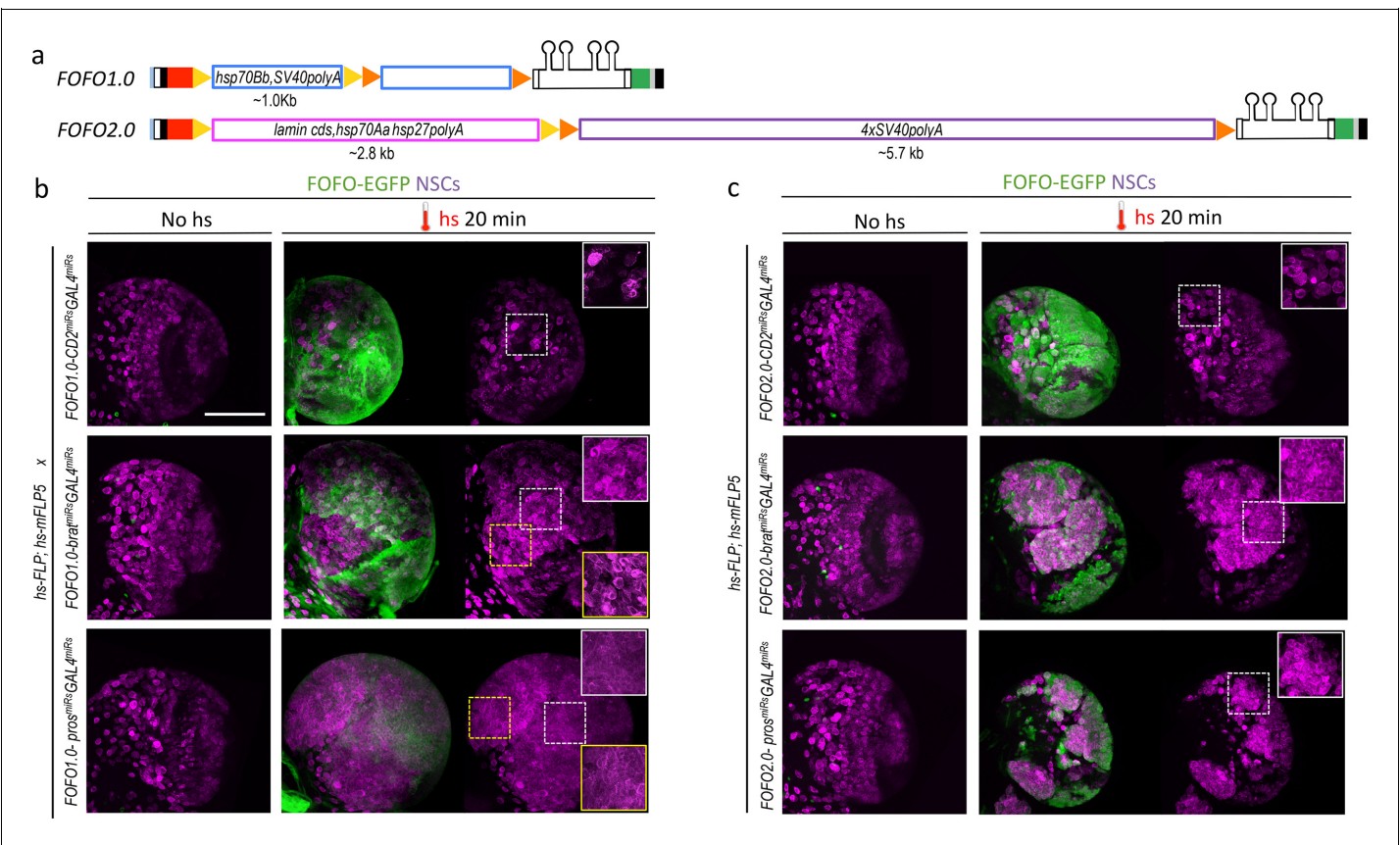

**Figure 2.** FOFO1.0 versus FOFO2.0. (**a**) FOFO.1.0 and FOFO2.0 differ in their STOP cassettes (drawn roughly to scale unlike remainder of construct); shapes are colour-coded as in *Figure 1a*. (**b**) Wandering third-instar larval brain lobes. In the absence of hs, the brains of animals carrying FOFO1.0 as well as hs-FLP1 and hs-mFLP5 look WT. Following hs, miR and EGFP expression is induced and supernumerary NSCs characteristic of these tumors are generated within the EGFP domain (notice NSC density in white-boxed insets). However, supernumerary NSCs outside the EGFP domain were also observed (notice NSC density in yellow-boxed inset, comparable to that of white-boxed inset of same sample). (**c**) Wandering third-instar larval brain lobes. In the absence of hs, the brains of animals carrying FOFO2.0 as well as hs-FLP1 and hs-mFLP5 look WT. Following hs, miR and EGFP expression is induced and supernumerary NSCs characteristic of these tumors are generated only within the EGFP domain (white-boxed insets). All images are maximum-intensity projections of Z-series but those of brains containing tumors are projections of only a few optical sections. Images are of a representative example obtained from two biological replicates (n > 10 per condition). Scale bar: 100 μm.

DOI: https://doi.org/10.7554/eLife.38393.004

The following source data and figure supplements are available for figure 2:

**Figure supplement 1.** FOFO2.0 precludes formation of supernumerary NSCs unless both FLP and mFLP5 are provided.

DOI: https://doi.org/10.7554/eLife.38393.005

**Figure supplement 1—source data 1.** Quantification of NSCs in indicated conditions.

DOI: https://doi.org/10.7554/eLife.38393.006

extensive patches of EGFP were observed in all transgenics (*FOFO1.0-CD2^{miRs}-GAL4^{miRs}*, *FOFO1.0-pros^{miRs}-GAL4^{miRs}* and *FOFO1.0-brat^{miRs}-GAL4^{miRs}*) (*Figure 2b*); occasional single cells labeled with EGFP could be seen in the absence of hs (average of 0.3 per brain lobe; n = 240 pooling data for FOFO1.0 and FOFO2.0 carrying *CD2^{miRs}*, *pros^{miRs}* or *brat^{miRs}* with no significant difference between genotypes). Furthermore, only in the presence of the oncogenic miRs were ectopic neural stem cells (NSCs) observed (*Figure 2b* white-boxed insets: note NSC density within EGFP patches in brains expressing oncogenic miRs versus controls). However, ectopic NSCs were sometimes observed also outside the EGFP domain in *FOFO1.0* carrying *pros^{miRs}* or *brat^{miRs}* (*Figure 2b* yellow-boxed inset). Because this was never seen in the absence of hs it was a Flippase-dependent process, likely due to inefficient termination of Pol II following excision of only one of the STOP cassettes. We concluded that our design, containing phenotype-inducing miRs ~ 200 bp downstream of STOP cassettes, was a sensitive reporter of Pol II readthrough (*Proudfoot, 2016*) and that this STOP cassette was unsuitable for our purpose.

We next generated a FOFO2.0 version containing two longer and potentially stronger, STOP cassettes: the Flybow one including *lamin* cds and a concatenation of four *SV40* terminators (*Jackson et al., 2001*; *Hadjieconomou et al., 2011*). As with FOFO1.0, in the presence of both hs-FLP and hs-mFLP5 and only after hs, extensive patches of EGFP were observed in all FOFO2.0 transgenics; EGFP single-cell labeling frequency was analogous to that for FOFO1.0; and only in the presence of oncogenic miRs were ectopic NSCs observed (*Figure 2c* white-boxed insets). This was the case for hs of 20 min and 1 hr. When we performed a double hs of 1.5 hr each 24 hr apart on *FOFO2.0-pros^{miRs}-GAL4^{miRs}* we occasionally saw tumors in the presence of only hs-mFLP5 (one central brain lineage in 8 out of 12 brains, which amounts to a frequency of ~0.3% as previously reported for cross-reactivity of hs-mFLP5 with FRT sites; *Hadjieconomou et al., 2011*). To ascertain that there was no leaky *miR* transcription in the absence of detectable EGFP, we counted the number of NSCs per larval central brain lobe and saw no differences between wild-type (WT) and *pros^{miRs}* and *brat^{miRs}* central brains, in the absence of hs or the presence of a single Flippase (or, in the few cases where hs-mFLP5 cross-reacted with FRT sites, outside the EGFP domain) (*Figure 2—figure supplement 1*). Crucially, with FOFO2.0 supernumerary NSCs were never observed outside the EGFP domain (*Figure 2c*). In summary, the FOFO2.0 design confirmed low-frequency cross-reactivity between mFLP5 and FRT sites but largely blocked miR transcription in the absence of either Flippase and successfully unblocked it in the presence of both, with perfect correspondence to EGFP reporter expression.

## Functionality of *GAL4^{miRs}*

To test efficacy of *GAL4^{miRs}*, we crossed *hs-FLP; hs-mFLP5,FOFO2.0-pros^{miRs}-GAL4^{miRs}* flies to those where all neural lineages are labeled in GAL4/UAS-dependent manner (GAL4 expressed in the domain of the Achaete-scute family transcription factor Asense (*Zhu et al., 2006*; *Bowman et al., 2008*) in the genotype *ase-GAL4,UAS-NLS::RFP*). The prediction was that wherever EGFP-labeled clones would be induced (by heat-shock) the RFP signal would be wiped out due to co-expression of *GAL4^{miRs}*. Indeed, following heat-shock, RFP-negative patches were observed in perfect overlap with EGFP-labeled clones, as expected from efficient GAL4 knock-down (*Figure 3*).

This experiment also illustrates successful combination of FLP/FOFO tools with GAL4/UAS as intended for independent genetic manipulations and genome-wide screens.

## New *enhancer-FLP(D)* transgenics

The next step was to employ FOFO2.0 to generate spatiotemporal controlled tumors in the larval CNS. Because of the report that mFLP5 can act on FRT sequences at low frequency but not the converse (*Hadjieconomou et al., 2011*), we used FLP for constitutive spatial control (lineage-specific *enhancer-FLP*) and mFLP5 for transiently induced temporal control (*hs-mFLP5*). Few lineage-specific FLP lines are currently available so we set out to generate some suited for our purpose. For type II lineages, we used the *R19H09* and *stg^{14}* enhancers previously described to be expressed therein (*Bayraktar et al., 2010*; *Wang et al., 2014*). We then browsed images reporting larval CNS expression of a large collection of *Drosophila* GAL4 lines (*Manning et al., 2012*) and selected 26 with restricted expression for further analysis. Induction of *pros* or *brat* tumors requires that these neural tumor suppressors be downregulated in progenitors, not in differentiated progeny. We thus

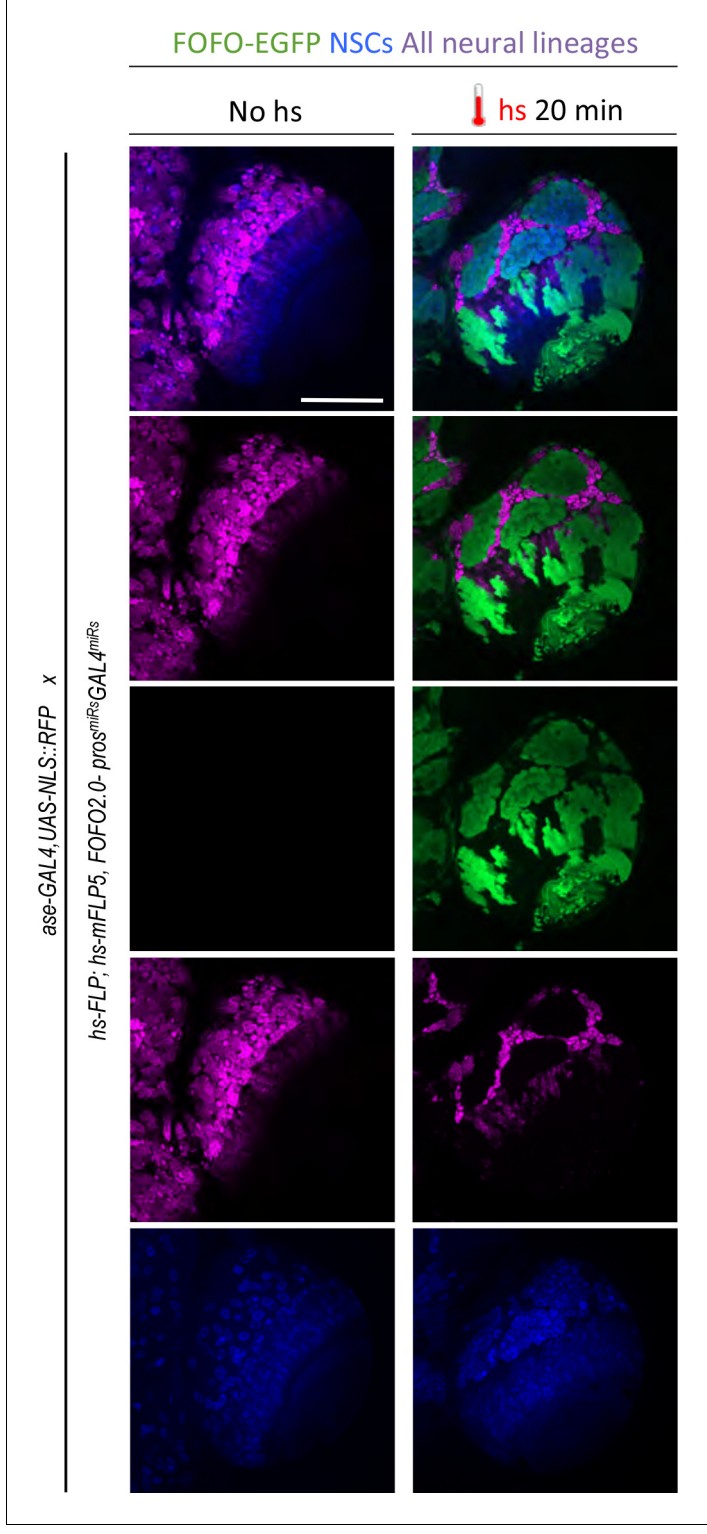

**Figure 3.** GAL4 miRs efficiently downregulate GAL4. *hs-FLP; hs-mFLP5,FOFO2.0-pros^{miRs}-GAL4^{miRs}* flies were crossed with *ase-GAL4,UAS-NLS::RFP* (which express RFP in all CNS lineages) flies. Wandering third-instar larval brain lobes of progeny are shown. Following heat-shock, EGFP and *GAL4^{miRs}* are expressed by the FOFO construct leading to RFP-negative patches in perfect overlap with EGFP-labeled clones as expected from efficient GAL4 knock-down. Images are of a representative example obtained from two biological replicates (n > 10 per condition). Scale bar: 100 μm.

DOI: https://doi.org/10.7554/eLife.38393.007

screened selected GAL4 lines for the ability to induce supernumerary NSCs (inferred by larger reporter gene domain) when crossed to *pros*[RNAi] – a functional screen for expression in neural progenitors. Ones of interest were further tested for the ability to induce supernumerary NSCs also when crossed to *brat*[RNAi]. Downregulation of *pros* should induce supernumerary NSCs in all central brain lineages (type I or II), whereas downregulation of *brat* should induce supernumerary NSCs only in type II. Furthermore, because we aimed to generate lines to induce an irreversible intrachromosomal recombination event, it was relevant to check not only expression at a particular timepoint but the 'complete' expression pattern from onset, permanently reported by a FLP-out event. Altogether, we chose nine enhancers from which to generate FLP lines (*Figure 4—figure supplement 1*).

Spatiotemporal control is constrained by the dynamics of the *enhancer-FLP*. The degree of reproducibility of FOFO-induced tumors depends on reproducibility of the expression domain of FLP, the strength of this expression and recombination efficiency. We employed a mutated form of FLP called FLP(D), which at position five contains an aspartic acid instead of glycine residue (*Babineau et al., 1985*) and is reported to be at least ten-fold more efficient than the original (*Nern et al., 2011*). Two different promoters were compared: that of *hsp70* and the *Drosophila* Synthetic Core Promoter (DSCP) employed in the generation of the GAL4 lines tested (*Pfeiffer et al., 2008*; *Han et al., 2011*). In all cases, expression controlled by the *hsp70* promoter was less widespread relative to that

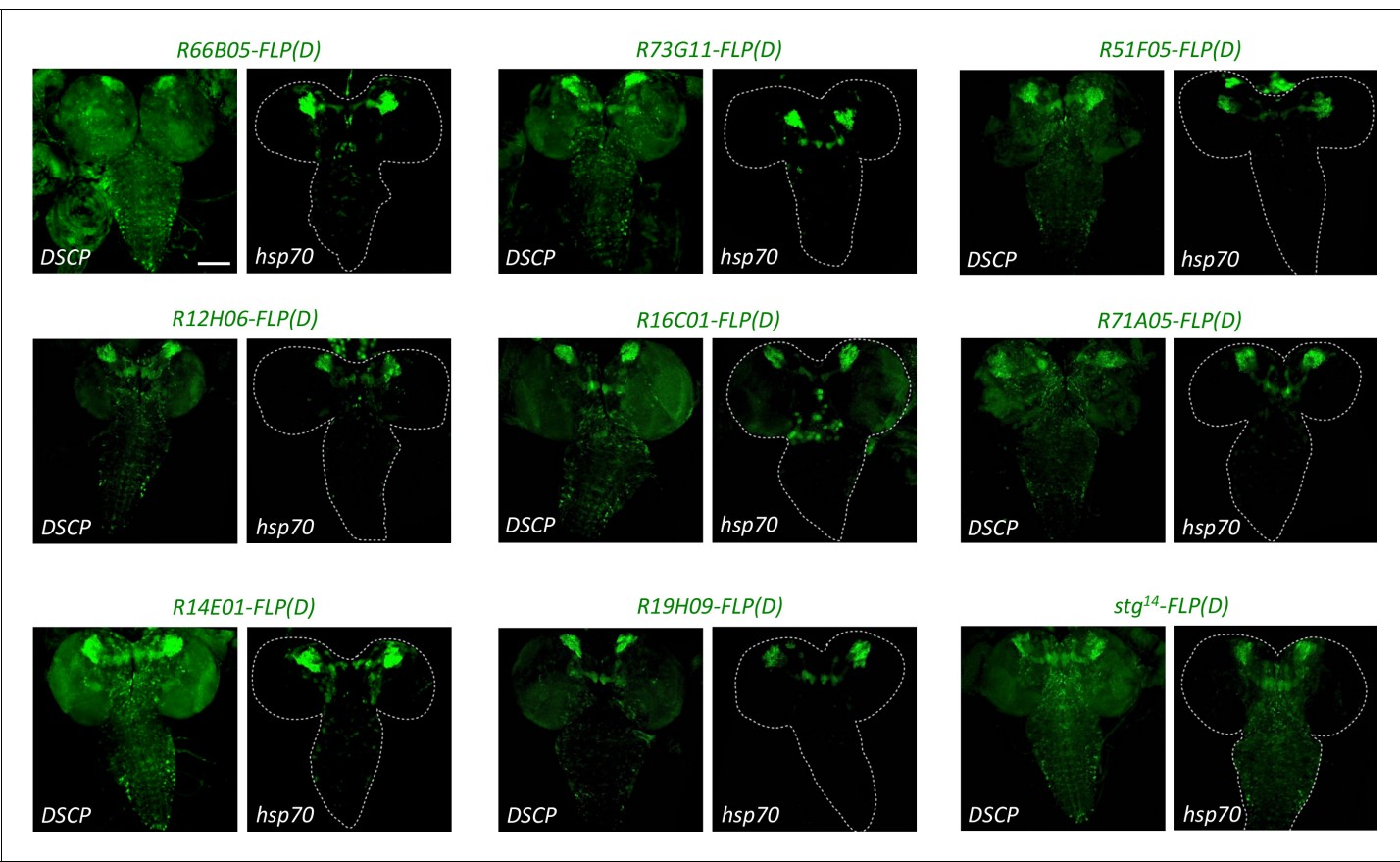

**Figure 4.** The *hsp70* promoter induces less expression of *enhancer-FLP(D)* lines than the *DSCP* promoter. New *enhancer-FLP(D)* lines were crossed to *act >STOP > GAL4,UAS-GFP* and wandering third-instar larval CNSs imaged for endogenous GFP expression. All genotypes were processed in parallel and imaged with identical conditions. In all cases, expression controlled by the *hsp70* promoter was less relative to that controlled by the DSCP, which could be due either to less background or sensitivity. Images are of a representative example obtained from two biological replicates (n > 10 per condition). Scale bar: 100 μm.
DOI: https://doi.org/10.7554/eLife.38393.008

The following figure supplement is available for figure 4:

**Figure supplement 1.** Examples of wandering third-instar larval CNSs of indicated genotypes.
DOI: https://doi.org/10.7554/eLife.38393.009

controlled by the DSCP (*Figure 4*), which could be due either to less background or sensitivity. Aiming for spatial restriction, we used the *hsp70* promoter lines for subsequent experiments.

## FLP cross-reactivity with mFRT71 at very low frequency

Newly generated *enhancer-FLP* lines containing the *hsp70* promoter were tested by crossing to *FOFO2.0-pros^{miRs}-GAL4^{miRs}*. The prediction was that no induction of supernumerary NSCs or EGFP expression would occur in progeny, whether or not heat-shocked, since *hs-mFLP5* was not provided. Most lines behaved as expected (no EGFP clusters containing supernumerary NSCs in the absence of hs: 0/33 for R14E01; 0/41 for R73G11; 0/31 for R19H09; 0/30 for R51F05; 0/36 for R71A05) but some *enhancer-FLPs* did very occasionally lead to induction of EGFP clusters containing supernumerary NSCs in the absence of hs (1/34 for R66B05; 5/39 for R12H06; 3/40 for R16C01; 4/28 for stg14); in all cases with a single spurious clone per brain. This indicates that FLP can cross-react with very low frequency with non-cognate mFRT71 sites (overall frequency of ~0.04% based on the number of such clones within the ~100 neural lineages per central brain; INPs were not included in this calculation, accounting for which would result in an even lower frequency). This cross-reactivity was never detected when crossing *hs-FLP* alone to *FOFO2.0* lines even following long double heat-shocks (*Figure 2— figure supplement 1*), suggesting that this phenomenon is either due to the FLP(D) structural variation, its enhanced recombination efficiency, and/or the fact that it is provided constitutively by the spatially restricted enhancers as opposed to transiently via a hs-mediated pulse. In any case, the almost negligible cross-reactivity indicated that these *enhancer-FLP* lines could be used for our purpose.

### *FOFO2.0*-induced tumor reproducibility

Each of the *FOFO2.0* transgenics (*FOFO2.0-CD2^{miRs}-GAL4^{miRs}*, *FOFO2.0-pros^{miRs}-GAL4^{miRs}* and *FOFO2.0-brat^{miRs}-GAL4^{miRs}*) was next recombined with *hs-mFLP5.* We then crossed these recombinants to *enhancer-FLP(D)* lines before combining them into a single stock. As expected, EGFP-labeled supernumerary NSCs were consistently observed following hs (*Figure 5a*). It was possible to combine all transgenes in a single animal stock with the exception of *R12H06-FLP(D)* and *FOFO2.0-pros^{miRs}-GAL4^{miRs}*, which was likely because this was the only one with a reasonable degree of tumor induction in the absence of heat-shock (*Figure 5b*). Following heat-shock, patches of EGFP-labeled supernumerary NSCs were observed for all enhancer-FLP(D) lines, with both *hs-mFLP5;FOFO2.0-CD2^{miRs}-GAL4^{miRs}* (controls) and *hs-mFLP5;FOFO2.0-pros^{miRs}-GAL4^{miRs}* but only the latter presented tumors (*Figure 5a*). Concerning reproducibility, in the first instance, we were looking for symmetry between brain lobes, suggestive of near-complete extent of recombination within the enhancer domain. The heat-shock regime that led to best tumor reproducibility in this regard was a double pulse of 1.5 hr each, with the first at the end of embryogenesis and a second during L1 (when brain NSCs are still quiescent), thus providing two doses of mFLP5 ~24 hr apart without intervening NSC divisions (*Figure 6*). We used *brat* tumors to test a number of conditions. With *hs-mFLP5;FOFO2.0-brat^{miRs}-GAL4^{miRs}* following heat-shock, patches of EGFP-labeled supernumerary NSCs were efficiently generated with *stg^{14}-FLP(D)* but rarely observed for *R19H09-FLP(D)* (*Figure 6a*), reflecting the different expression dynamics of the two enhancers. We then compared reproducibility dependence on different recombinase loading regimes: 2 versus 1 hs; and 2 versus 1 copy of enhancer-FLP(D). Tumors were largest when cells were delivered double-loads of each of the recombinases (2 hs in homozygous enhancer-FLP(D) animals) and smallest when a single dose of each recombinase was provided (*Figure 6b*). The double-load of mFLP5 and of FLP(D) greatly reduced tumor asymmetry between lobes (*Figure 6c*). In summary we were able to generate spatiotemporally controlled lineage-restricted labeled CNS tumors in a single stock in the absence of the GAL4/UAS system.

## Discussion

We engineered genetic tools with which to generate labeled lineage-restricted CNS tumors (applicable to any other deleterious genetic perturbation) in a single stock, and independently of GAL4/UAS. We demonstrate successful combination of novel FLP/FOFO tools with GAL4/UAS and efficacious GAL4 knock-down within domains of FLP/mFLP5 and GAL4 intersection. This validates our tool for independent genetic manipulations in strictly non-overlapping domains, which is transformative

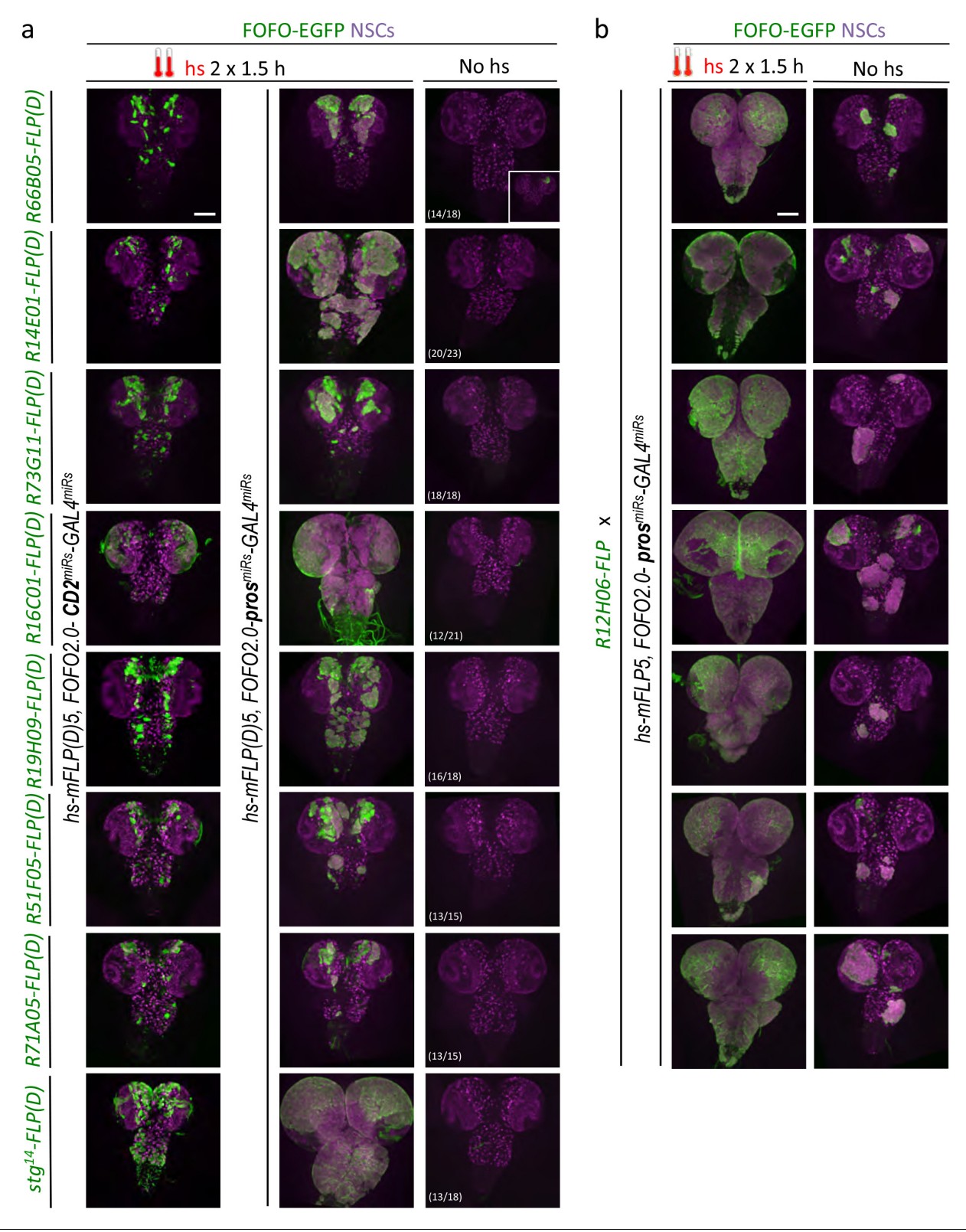

**Figure 5.** FOFO2.0-mediated lineage-restricted CNS tumor generation within a single stock. (a) Wandering third-instar CNSs of hs-induced labeled tumors obtained with eight *enhancer-FLP(D)* and *hs-mFLP5,FOFO2.0-pros^{miRs}-GAL4^{miRs}* compared with non-tumor-labeled lineages (same *enhancer-FLP(D)s* with *hs-mFLP5,FOFO2.0-CD2^{miRs}-GAL4^{miRs}*) and background (no hs) tumor incidence. In the absence of heat-shock, tumors were occasionally induced with incomplete penetrance (inset in top right; numbers indicate frequency of CNSs devoid of tumours) but these were much smaller than

*Figure 5 continued on next page*

*Figure 5 continued*

those intentionally induced by heat-shock. (b) Wandering third-instar larval CNSs from progeny of the cross between indicated genotypes. When subject to heat-shock, extensive tumors are induced throughout the CNS (labeled in green and containing supernumerary NSCs). In the absence of heat-shock, tumors (albeit much smaller) are induced. (a–b) All images are maximum-intensity projections of Z-series; obtained from two biological replicates (n > 10 per condition and exact number indicated for the background condition in a – third column). Scale bar: 100 µm.
DOI: https://doi.org/10.7554/eLife.38393.010

for the study of cell non-autonomous effects. Our design opens up for the first time the ability to perform efficient genome-wide screening for non-autonomous effects on deleterious genotypes.

We show that employment of 4 miRs is efficacious and permits simultaneous downregulation of multiple genes in the labeled domain; furthermore, T2A sequences can be added for simultaneous overexpression of coding sequences in addition to that for a reporter. The system can be used also to refine spatial domains, intersecting various enhancer-recombinases (in addition or not to hs control).

The sensitivity of our design (with miR expression inducing a readily detectable and quantifiable phenotype even in non-labeled cells) allowed us to define STOP cassettes appropriate to curb even short Pol II readthrough. The discrete number of progenitors from which tumors are initiated provided a convenient platform to quantify Flippase cross-reaction and revealed low-level cross-reaction of FLP(D) with mFRT71 sites, not described before. The degree of tumor reproducibility reported differences in expression dynamics of the lineage-restricted enhancers (e.g. as seen by more asymmetric *brat* tumors with *R19H09-FLP(D)* than with *stg$^{14}$-FLP(D)*) and incomplete extent of recombination within the enhancer domain. Reproducibility could be improved by increased loading of recombinases so in cases where reproducibility is desired we recommend using multiple copies of recombinase transgenes.

With this setup, any desired GAL4 line can now be added to the stock containing the other elements (spatially restricted-FLP, hs-mFLP5, FOFO) and screens can be performed with a number of convenient criteria. For example, the presence of larval neural tumors induces developmental delay whose extent is proportional to tumor size (our unpublished observation); and in some lineages leads to adult sub-lethality (i.e. presence of adults bearing tumors in a sub-Mendelian proportion). Therefore, the extent of developmental delay and of adult escapers can be used as first-pass proxies for tumor size, for speedy screening of non cell-autonomous modifiers of these parameters. Tumor volume can be subsequently measured directly. Additionally, a FOFO version containing a Luciferase reporter can be generated in order to use Luciferase activity as an efficient method of quantifying reporter-expressing cells (in our case tumor volume) in homogenized tissue (*Homem et al., 2014*).

Custom-made FOFO tools can be applied to any desired topic and cell types. Control flies (those with miRs against CD2) will be available 'off the shelf' and experimental ones can be generated by either gene synthesis or modification of the control plasmid; or by CRISPR-modification of control host flies. It would be interesting to compare efficacy of these strategies as host flies could contain already other modules of interest. Other recombinase pairs can also be employed where mFLP5/FLP cross-reactivity is a concern. Within the CNS, other applications include investigating cell non-autonomous modifications of axon misguidance, perturbed arbor growth or synapse formation, roles of glia on neurodegeneration, etc. Furthermore, even without gene perturbations, the FOFO tool allows sparse labeling of specifically targeted cells (sparseness achieved by short heat-shock and cell-type targeting provided by *enhancer-FLP*), which is extremely useful for studying cellular morphology and/or migration. Beyond the CNS, the resurgence of interest in metabolism and physiology, for example, has had strong contribution from *Drosophila* research (*Rajan and Perrimon, 2013*). These are disciplines that involve interplay between cell types and different organs and tools like the ones described here will undoubtedly propel them forward.

The principles of the FOFO design can be applied to other model organisms where distinct site-specific recombinases work, such as is the case for zebrafish and mouse (*Nern et al., 2011*; *Olorunniji et al., 2016*; *Carney and Mosimann, 2018*; *Yoshimura et al., 2018*) for refined spatial and/or temporal control of gene expression. In zebrafish, heat-shock induced gene expression allows for faster and/or focal induction of gene expression as compared to drug-induced expression (*Halloran et al., 2000*). Direct translation of a FOFO tool with the aim here described (large-scale screening for non-autonomous effects) is feasible in zebrafish by employment of the GAL4/UAS or

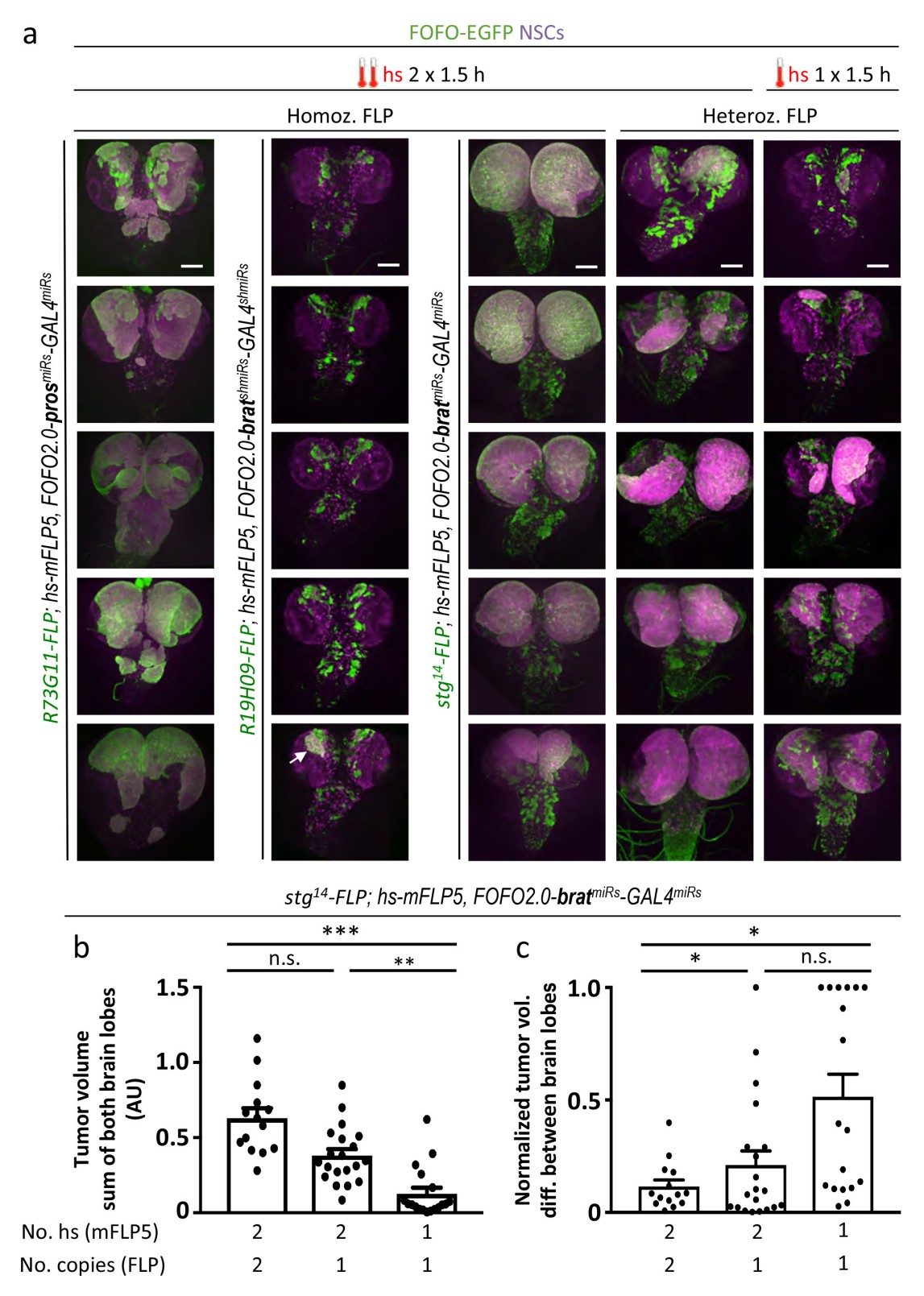

**Figure 6.** Selection of appropriate enhancer-FLP(D) in combination with hs-mFLP5 allows reproducible CNS tumor generation within a single stock via FOFO2.0. (a) Representative images of white prepupal CNSs in which the stated enhancer-FLP was employed as either homozygous or heterozygous as indicated, with *pros^miRs* or *brat^miRs* with the hs regimes indicated (arrow points at rare tumor generated with *R19H09-FLP*). All images are maximum

*Figure 6 continued on next page*

*Figure 6 continued*

intensity projections of Z-series. Scale bar: 100 µm. (b) Quantification of EGFP volumes of *brat^miRs^* tumors. (c) Normalized tumor volume differences between brain lobes. * p < 0.05; ** p < 0.01; *** p < 0.001; n.s. = not significant.

DOI: https://doi.org/10.7554/eLife.38393.011

The following source data is available for figure 6:

**Source data 1.** Quantification of tumor and brain lobe volumes.

DOI: https://doi.org/10.7554/eLife.38393.012

Q/QUAS systems (*Subedi et al., 2014*; *Kawakami et al., 2016*). In mouse, one-way thermal shock can be focally-induced (thus minimizing unwanted damage of most cells) is by Brownian motion of iron oxide nanoparticles when subject to a magnetic field. Once injected into specific tissues, these nanoparticles remain static and can be visualized by magnetic resonance imaging (*Pankhurst et al., 2003*), which means the site of injection, and therefore of heat-shock, can be located any time post-injection. Translating the example of this study into mice, induction of tumorigenesis focally in specific cell types by a combination of heat-shock and a cell-type-specific recombinase, in a way that allows identification of exactly where the tumor was initiated, will be invaluable to study the earliest events in mammalian tumorigenesis. This is largely a 'black box' in in vivo mammalian cancer studies, with assumed extrapolation from in vitro findings, since by the time a tumor can be visualized it is usually already of a substantially advanced stage. FOFO applications are thus myriad and versatile.

## Materials and methods

### Plasmid backbone

A modified *pCaSpeR* plasmid containing an *actin5C* promoter and a PhiC31-Integrase *attB* site was kindly provided by C. Alexandre and further modified as described next. To enhance expression and avoid positional effects, gypsy insulators were amplified from *pVALIUM20*[24] adding 5' EcoRI and XhoI, and 3' BamHI and NheI restriction sites: the gypsy PCR product digested with EcoRI and NheI was cloned into identical sites in the modified *pCaSpeR*, making *act5C-gypsy1*; the *gypsy* PCR product digested with XhoI and BamHI was cloned into identical sites in *act5C-gypsy1*, making *act5C-gypsy2*. To minimise recombination, this plasmid as well as its FOFO derivatives were best grown in XL10-Gold Ultracompetent Cells (Agilent Technologies, Cat. No. 200314) at 30°C at 150 rpm.

### FOFO modules

An initial FOFO insert containing *CD2^miRs^-GAL4^miRs^* and restriction sites at key locations for modularity was generated by gene synthesis (Integrated DNA Technologies) and cloned into XhoI-NotI sites in *act5C-gypsy2*.

### Short hairpin design and exchange

All miR sequences were embedded in the *Drosophila miR-1* stem-loop backbone (*Haley et al., 2008*), within the *ftz* intron (*Haley et al., 2010*). Control miRs were those previously used to downregulate CD2 (*Yu et al., 2009*); both *GAL4 miRs* and one each for *pros* and *brat* were sequences selected by the Transgenic RNAi Project (TRiP; *Ni et al., 2011*); other *pros* and *brat miRs* were selected by us (sequences below) from the output of the Designer of siRNA (DSIR) software (*Vert et al., 2011*; http://biodev.extra.cea.fr/DSIR/DSIR.html). In brief, target mRNA sequences were fed into the software and output sequences BLASTed against the *Drosophila* transcriptome; sequences with ≥16 bp contiguous matches to other targets were excluded. Hairpin sequences targeting *pros* or *brat* along with ones targeting *GAL4*, flanked by AscI on the 5' end and AvrII on the 3' end, were generated by gene synthesis (GenScript). The AscI-AvrII fragments were cloned into identical sites in FOFO1.0, making *FOFO1.0-pros^miRs^-GAL4^miRs^* or *FOFO1.0-brat^miRs^-GAL4^miRs^*. The restriction sites (lowercase) and hairpin sequences (sense and antisense indicated in **bold**) used in this study were:

GAL4$^{miRs}$

cctaggAACATCCCATAAAACATCCCATATTCAGCCGCTAGCAGT**CAGGATTATTTGTACAAGATA**
**TAGTTATATT**CAAGCATA**TATCTTGTACAAATAATCCTG**GCGAATTCAGGCGAGACATCGGAG
TTGAAACTAAAACTGAAATTTACTAGAAAACATCCCATAAAACATCCCATATTCAGCCGCTAGCAG
T**TCGGAAGAGAGTAGTAACAAA**TAGTTATATTCAAGCATA**TTTGTTACTACTCTCTTCCGA**GCGAA
TTCAGGCGAGACATCGGAGTTGAAACTAAAACTGAAATTTCCTAGG

pros$^{miRs}$

ggcgcgccAACATCCCATAAAACATCCCATATTCAGCCGCTAGCAGT**CAGGATGTGGAACAAGAA-**
**CAA**TAGTTATATTCAAGCATA**TTGTTCTTGTTCCACATCCTG**GCGAATTCAGGCGAGACATCGGAG
TTGAAACTAAAACTGAAATTTACTAGAAAACATCCCATAAAACATCCCATATTCAGCCGCTAGCAG
T**TAGCAGTAGTAGTAACAATAA**TAGTTATATTCAAGCATA**TTATTGTTACTACTACTGCTA**GCGAA
TTCAGGCGAGACATCGGAGTTGAAACTAAAACTGAAATTTCCTAGG

brat$^{miRs}$

ggcgcgccAACATCCCATAAAACATCCCATATTCAGCCGCTAGCAGT**CTGTGTCAAGGTGTTCAAC**
**TA**TAGTTATATTCAAGCATA**TAGTTGAACACCTTGACACAG**GCGAATTCAGGCGAGACATCGGAG
TTGAAACTAAAACTGAAATTTACTAGAAAACATCCCATAAAACATCCCATATTCAGCCGCTAGCAG
T**CGGCGTGGTGGTCAACGACAA**TAGTTATATTCAAGCATA**TTGTCGTTGACCACCACGCCG**GC-
GAATTCAGGCGAGACATCGGAGTTGAAACTAAAACTGAAATTTCCTAGG

## STOP cassettes

FOFO1.0 contained two identical STOP cassettes consisting of *hsp70Bb* (*Nern et al., 2011*) and *SV40* terminators. FOFO2.0 contained a first STOP cassette consisting of the *lamin cds* plus *hsp70Aa* and *hsp27 polyA* generated by PCR using FB2.0 (*Hadjieconomou et al., 2011*) as template with the primers (Forward and Reverse always indicated in this order): gat cga tcc ccg ggt acc gcg gcc gcA TAG GGA ATT GGG AAT TCG C and cga att ccc aat tcc cgt tta aaC TCG AGG GTA CCA GAT CTG (uppercase indicating complementarity to template); and a second STOP cassette consisting of four tandem *SV40 polyA* sequences generated by PCR using the plasmid *Lox-Stop-Lox TOPO* (Addgene; *Jackson et al., 2001*) as template with the primers: gat cga tcc ccg ggt acc gcg gcc gcG AAG TTC CTA TAC TTT CTA G and ttt ggc ttt agt cga CTC TAG TTT AGG CGT AAT CG. Products were inserted by Gibson Assembly (NEB) into *FOFO-EGFPnls* backbones digested with NotI and PmeI to remove the existing STOP cassettes. Primers were designed either manually or, for Gibson Assembly, with the New England Biolabs builder tool (http://nebuilder.neb.com/).

## Reporter

The reporter gene used was EGFP, fused in its N-terminal to a membrane targeting sequence (CD8), obtained by PCR from FB2.0 (*Hadjieconomou et al., 2011*); or in its C-terminal to the SV40 NLS GSPPKKKRKVEDV (GGA TCC CCC CCC AAG AAG AAG CGC AAG GTG GAG GAC GTC TAG) engineered by Gibson Assembly (New England Biolabs) from a sequence kindly provided by G. Struhl and including a Kozak consensus. The 3'UTR used was *His2av3'UTR*-PolyA (*Manning et al., 2012*).

## Enhancer-FLPs

For the *enhancer-FLP(D)* constructs, the plasmid *pDEST-HemmarG* (Addgene; *Han et al., 2011*) was modified using Gibson Assembly (New England Biolabs) as described next. *CD4-tdGFP cds* was removed with XhoI and XbaI and replaced by a PCR fragment encoding FLP(D) obtained from *pJFRC150-20XUAS-IVS-Flp1::PEST* (Addgene; *Nern et al., 2011*) with the primers: cct ttt cgt tta gcc aag act cga gAA TCA AAA TGC CGC AGT TTG and act ggc tta gtt aat taa ttc tag att aAA TAC GGC GAT TGA TGT AG. We call the resulting plasmid *pDEST-Hemmar-FLP(D)*. This was transformed into One Shot ccdB Survival 2 T1R Competent Cells (Life Technologies, Cat. No. A10460). A modified version of *pDEST-Hemmar-FLP(D)* containing the *DSCP* promoter (*Pfeiffer et al., 2008*) and the *ftz* intron (*Haley et al., 2010*) was generated using Gibson Assembly (New England Biolabs) using *pBPGUw* as a template. *pDEST-HemmarG* was digested with BbvcI and XbaI, removing part of the *ccdB cds* as well as the *hsp70* promoter, the *zeste* intron and *CD4-tdGFP cds*. PCR fragments

containing the sequences for completing the *ccdB cds* as well as for the *DSCP* promoter, *ftz* intron and *FLP(D) cds* were obtained using the primers: gga aaa tca gga agg gat ggc tga ggT CGC CCG GTT TAT TGA AAT G and cgg cca att cAG CTG AAC GAG AAA CGT AAA ATG (*attR1 +ccdB cds*), tcg ttc agc tGA ATT GGC CGC GTT TAA AC and gat tct cga gCC TGC AGG TCT TTG CAA TC (*DSCP* and *ftz* intron), gac ctg cag gCT CGA GAA TCA AAA TGC C and act ggc tta gtt aat taa ttc tag atc tag att aAA TAC GGC GAT TGA TGT AG (*FLP(D) cds)* and assembled into the BbvcI-XbaI *pDEST-HemmarG* fragment. We call the resulting plasmid *pDEST-Hemmar-DSCP-ftz-FLP(D)*.

Enhancer fragments were generated by PCR from gDNA and cloned into *pENTR/D-TOPO* (Life Technologies, Cat. No. K2400-20). Primer sequences contained CACC at the 5' end of the forward primer for Gateway cloning. LR reaction products between *pENTR/D-TOPO* containing enhancer fragments and *pDEST-Hemmar-FLP(D)* or *pDEST-Hemmar-DSCP-ftz-FLP(D)* were used to generate transgenic flies.

### *Drosophila* stocks and transgenesis

*hs-FLP, UAS-CD8::GFP, UAS-brat^{SH}, UAS-pros^{SH}/CyO* and Janelia Farm GAL4 lines were obtained from the Bloomington Stock Centre; *act >STOP > GAL4,UAS-GFP* was a gift from W. Chia; *UAS-FLP,tub >STOP > GAL4,UAS-CD8::GFP* was a gift from M. Landgraf; *ase-GAL4* recombined with *UAS-myr::RFP* was a gift from A. Bailey. *Bc/CyO; hs-mFLP5/TM2* was a gift from I. Salecker.

PhiC31 Integrase-mediated transgenesis was performed by BestGene Inc. into *attP40* (*FOFO*), *attP18* (*enhancer-FLP*), *attP16* (*hs-FLP* or *hs-mFLP5*) strains mutant for the gene *white*, which results in white eyes; since all transgenes included the *white* gene, insertions were selected by eye color in the F1 generation. For *FOFO* transgenesis, animals were injected and reared at 18°C.

### Heat-shocks

Larvae were heat-shocked by tube emersion into a 37°C water-bath. Duration as indicated in text and/or figures.

### Immunohistochemistry and imaging

For immunohistochemistry, CNSs were fixed for 15 min in 3.7% formaldehyde in PBS. Mouse anti-Miranda (mAb81 1/50; gift from F. Matsuzaki) was used to label NSCs. Secondary antibodies were conjugated to either Alexa-Fluor-488 or Alexa-Fluor-555 (Molecular Probes) and used at 1/500. Tissues were mounted in Vectashield (Vector Laboratories) and images obtained using a Zeiss LSM510 confocal microscope. Images were acquired using the same confocal (laser power, gain and pinhole) conditions. Maximum intensity z-stack projections were generated and brightness/contrast of whole images adjusted with FIJI software.

### Quantifications and statistics

Neither randomization nor blinding was used except for data shown in *Figure 2—figure supplement 1* and *Figure 6b–c*, where NSC counts and tumour volume measurements were performed blind for genotype. Here, each data point corresponds to a different individual of the designated genotype or condition. Sample size calculation is unwarranted due to the small standard deviation of the number of NSCs per central brain lobe in WT and the large effect that tumour induction has on this (many standard deviations above the mean). Data was checked for normalcy via the Liliefors test; significance of difference between each genotype and WT was tested by Ordinary One Way ANOVA, multiple comparisons. For *Figure 6*, a complete Z-stack was acquired for every brain (both lobes). Quantification of tumor volume in each lobe was performed with Amira-Avizo Software (Thermo Scientific) using overlapping EGFP and anti-Miranda to identify tumors. Here, tumors were traced throughout the Z-stack to generate the volume of the traced tumour using the segmentation tool in the software package. Tumor volume of each animal was obtained by summing the volume of both brain lobes. The proportional difference of tumor volumes between brain lobes of each animal was obtained by subtracting the smaller volume (S) from the bigger volume (B) and dividing this by the sum of the two (S + B), that is (B-S)/(B + S). Significance of difference between each condition was tested by Kruskal-Wallis with Dunn's multiple comparisons test as post hoc analysis. Each experiment was performed twice (biological replicates). Biological replicates refer to biologically distinct samples (independent crosses) grown in the same conditions and undergone the experimental

procedure; sample number is indicated in each appropriate figure legend. No data was excluded. Statistical tests and graphs were generated using Prism software.

### Reagent availability

Plasmids and transgenic flies are deposited in stock centres. Sequence of pFOFO2.0-CD2$^{miRs}$-GAL4-$^{miRs}$-EGFPnls is provided as *Supplementary File 1*.

## Acknowledgements

We thank C. Alexandre, A Bailey, W Chia, B Haley, M Landgraf, I Salecker and G Struhl for reagents or advice; and T Carter, K Chester, J Clarke, M Fanto, TE Rusten, and D Schmucker for helpful comments on the manuscript. We are also grateful to N Carvajal, MJ Cruz and A Miedzik and for technical assistance with some experiments.

## Additional information

### Funding

| Funder | Grant reference number | Author |
| --- | --- | --- |
| Cancer Research UK | Career Development Fellowship to Rita Sousa-Nunes | Rita Sousa-Nunes |

The funders had no role in study design, data collection and interpretation, or the decision to submit the work for publication.

### Author contributions

Andrea Chai, Data curation, Formal analysis, Validation, Investigation, Visualization, Writing—review and editing; Ana M Mateus, Data curation, Validation, Investigation, Visualization, Methodology, Writing—review and editing; Fazal Oozeer, Data curation, Investigation, Methodology; Rita Sousa-Nunes, Conceptualization, Resources, Supervision, Funding acquisition, Methodology, Writing—original draft, Project administration, Writing—review and editing

### Author ORCIDs

Rita Sousa-Nunes (iD) http://orcid.org/0000-0002-7401-8081

### Decision letter and Author response

Decision letter https://doi.org/10.7554/eLife.38393.016
Author response https://doi.org/10.7554/eLife.38393.017

## Additional files

### Supplementary files

• Supplementary file 1. Sequence of pFOFO2.0-CD2miRs-GAL4miRs-EGFPnls.
DOI: https://doi.org/10.7554/eLife.38393.013
• Transparent reporting form
DOI: https://doi.org/10.7554/eLife.38393.014

### Data availability

All data generated or analysed during this study are included in the manuscript and supporting files. Source data files have been provided for Figure 2-supplement figure 1 and for Figure 6; the FOFO2.0 plasmid sequence has been uploaded; plasmids have been deposited in Addgene (https://www.addgene.org/depositing/76160/) and transgenic flies have been deposited in the Bloomington Drosophila Stock Center.

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
