## [Decision Letter]

Thank you for submitting your article "Spatiotemporally controlled genetic perturbation for efficient large-scale studies of cell non-autonomous effects" for consideration by *eLife*. Your article has been reviewed by three peer reviewers, and the evaluation has been overseen by Hugo Bellen as the Reviewing Editor, and Didier Stainier as the Senior Editor. The following individuals involved in review of your submission have agreed to reveal their identity: Chris Doe (Reviewer #1); Marc Freeman (Reviewer #2).

The reviewers have discussed the reviews with one another and the Reviewing Editor has drafted this decision to help you prepare a revised submission.

Summary:

This manuscript describes a genetic toolkit that can be used to generate a specific phenotype in a targeted patch of cells (which I will call the screening phenotype) while concurrently performing gene knock-down/misexpression in the adjacent cells to detect non-cell autonomous modification of the screening phenotype. The screening phenotype transgene is kept inactive until spatial and temporal cues lead to 'flip out' of stop codons, thereby allowing expression of the transgene and initiation of the screening phenotype.

Chai et al. describe this novel method for temporal and spatial control of genetic perturbation using two flip-out cassettes that can be regulated by different transgenes. The method only uses the FLP FRT system and leaves other genetic tools such as other binary expression systems free to use for further genetic manipulation of samples. The authors included a miRNA targeting GAL4 in their construct to silence GAL4 in the expression domain. This allows cell autonomous and non-cell autonomous effects to be separated. Moreover, all the components to induce the expression of the construct can be included in a single parental stock, enabling crossing a single stock to effectors such as RNAi libraries to conduct F1 screens. The authors generate multiple versions of this construct to knock down tumor suppressors and GAL4. They generated enhancer FLPase stocks that can be used for neuronal expression and optimized stop cassettes, flippase constructs and knock down constructs separately. Overall this is a sophisticated method that can address niche questions quite nicely. I believe the method will be very useful for the laboratories that aim to address such questions. This elegant system provides a new tool for *Drosophila* researchers interested in performing screens for non-cell autonomous functions. The idea and execution of this system are quite impressive, yet we have some concerns about the general utility of the system described below.

Essential revisions:

1) Using the system 'off the shelf' is a stated goal of the authors, and an admirable one. But the system is designed to create just one, very specific screening phenotype: neuroblast tumor formation. This is because the screening transgene has miR against prospero or brat only. Generating other screening phenotypes would require significant molecular genetics to adapt the system to create different phenotypes. As it stands, this tool is only useful for research into non-cell autonomous modifiers of neuroblast-derived tumors.

2) Using the system 'off the shelf' would also be limited to a single Gal4 line, named *enhancer2-gal4*, as described in Figure 1B. Subsequent figures show crosses to different Gal4 lines, but I don't understand how that works if the gal4 is part of the parental screening fly as shown in the schematic in Figure 1.

3) I have some concern about 'leaky' expression of the screening phenotype, due to unwanted FLP/mFLP5 recombination. The number of events per lineage is calculated, but since there are 100 neuroblasts and 1000 or more INPs in the brain, a low frequency per lineage may translate into an unacceptable level of events per brain. Could the authors give the tumor frequency per brain for each genotype?

4) Figure 2A should have a key for the colored boxes: what do each represent?

Also, in Figure 2 there is small number of green cells that can be seen in both FOFO1 and FOFO2 non heat shocked samples. These do not seem to cause a problem since tumors only form when pros or brat are removed in progenitors. For other applications this leakiness (which most likely results from heat shock promoters leakiness) can be problematic. This should be discussed and quantified.

5) The authors aim to obtain reproducibility of tumor induction but never really compare the tumors in different animals. Is the increase in number of NSC similar between samples induced the same way? How does it change when heat shock regime is changed? These should be quantified.

6) The temporal control of expression is inherently dependent on the expression dynamics of enhancers. Judging from Figure 4—figure supplement 1, almost all the enhancers that the authors use unlock the construct in a larger set of cells that end up expressing the enhancer at third instar (comparing enhancer-*GAL4 Tub>Stop>GAL4*, UAS-FLP mediated GFP expression to direct enhancer-GAL4 mediated expression). Therefore both the temporal control and special control is limited. The wording in the text should be toned down when it comes to spatial and temporal control of expression and caveats should be more clearly stated.

7) The heat shocks are conducted so early for tumorigenesis and conducting two heat shocks completely ablates the temporal control aspect in the experiments. Is this tumor induction method better/more robust than a single flip out regulated by enhancer specific Flp? A head to head comparison should be included.

8) The authors should deposit their plasmids in Addgene or BDGRC and explain how to expand applications for different applications by customizing their tools. They should also deposit their transgenic flies in the BDSC. They nowhere mention that they are planning or willing to do this. These reagents and stocks should be deposited before the paper is accepted or at the time the paper is going to press.

[Editors' note: further revisions were requested prior to acceptance, as described below.]

Thank you for resubmitting your work entitled "Spatiotemporally controlled genetic perturbation for efficient large-scale studies of cell non-autonomous effects" for further consideration at *eLife*. Your revised article has been favorably evaluated by three reviewers, Didier Stainier as the Senior Editor, and Hugo Bellen as the Reviewing Editor.

The manuscript has been improved but there are some remaining issues that need to be addressed before acceptance, as outlined below:

One of the reviewers wrote the following:

The Abstract and end of the Introduction still makes it look like this method can be used "off the shelf" when in fact any different application from their example requires genome engineering to replace their example microRNAs with different microRNAs or misexpression ORFs. So I would like to see the Abstract changed from "Altogether, our design opens up efficient genome-wide screens on any deleterious phenotype." to "Altogether, our design opens up efficient genome-wide screens on any deleterious phenotype, once genome engineering is used to place the desired miRNA or ORF into our genotype."

We agree that this change should be implemented.

Also, we want to ensure that all the reagents will be available from the BDSC without MTA at publication time.

---

## [Author Response]

Essential revisions:1) Using the system 'off the shelf' is a stated goal of the authors, and an admirable one. But the system is designed to create just one, very specific screening phenotype: neuroblast tumor formation. This is because the screening transgene has miR against prospero or brat only. Generating other screening phenotypes would require significant molecular genetics to adapt the system to create different phenotypes. As it stands, this tool is only useful for research into non-cell autonomous modifiers of neuroblast-derived tumors.

What we did was to conceive and implement a system to be used widely, but that indeed has to be tailored in the particular module downstream of the STOPs for a specific screening phenotype. As detailed in the Summary, this involved optimization of a number of features, which resulted in a “template design” that will benefit the community going forward. Depending on their biological question, each researcher can adapt the system for their own purposes (designing their own module downstream of the STOPs). We stated that (Discussion) “Custom-made FOFO tools can be applied to any desired topic and cell types.” We have now added: “Control flies (those with miRs against CD2) will be available ‘off the shelf’ and experimental ones can be generated by either gene synthesis or modification of the control plasmid; or by CRISPR-modification of control host flies. It would be interesting to compare efficacy of these strategies as host flies could contain already other modules of interest.”

2) Using the system 'off the shelf' would also be limited to a single Gal4 line, named enhancer2-gal4, as described in Figure 1B. Subsequent figures show crosses to different Gal4 lines, but I don't understand how that works if the gal4 is part of the parental screening fly as shown in the schematic in Figure 1.

Indeed, the intention is to combine in a single stock the FOFO-containing strain with appropriate recombinases and GAL4 transgenes (single or multiple GAL4 transgenes, or split GAL4 modules) dependent on each researcher’s goal (as per the parental fly shown on the left of Figure 1B). The only figure in which we show an example of FOFO and GAL4 combination is Figure 3; here, the aim was to test the miRs against GAL4 rather than the generation of a screening fly so instead of combining FOFO and GAL4 in the same stock we crossed the two genotypes (indicated in separate lines on the left of the figure with an ‘x’ between them) as that served our purpose.

3) I have some concern about 'leaky' expression of the screening phenotype, due to unwanted FLP/mFLP5 recombination. The number of events per lineage is calculated, but since there are 100 neuroblasts and 1000 or more INPs in the brain, a low frequency per lineage may translate into an unacceptable level of events per brain. Could the authors give the tumor frequency per brain for each genotype?

Two tests were called-for in this respect, which we performed:

i) FOFO tumor frequency in the presence of hs-mFLP5 and absence of enhancer-FLP;

ii) FOFO tumor frequency in the presence of enhancer-FLP and absence of hs-mFLP5.

i) Consisted of testing a single genotype and the frequency was given in the last paragraph of the subsection “Efficacy of STOP cassettes”.

ii) Consisted of testing 9 genotypes (corresponding to the 9 different enhancerFLPs) and had been given only as the overall frequency. We have now added the numbers obtained for each of the 5 genotypes in which we did not observe cross-reactivity and the 4 in which we did. We state: “Most lines behaved as expected (no EGFP clusters containing supernumerary NSCs in the absence of hs: 0/33 for R14E01; 0/41 for R73G11; 0/31 for R19H09; 0/30 for R51F05; 0/36 for R71A05) but some *enhancer-FLPs* did very occasionally lead to induction of EGFP clusters containing supernumerary NSCs in the absence of hs (1/34 for R66B05; 5/39 for R12H06; 3/40 for R16C01; 4/28 for stg14); in all cases with a single spurious clone per brain.” The combined frequency was therefore 13/31,200 NSCs (9 genotypes pooled), which amounts to ~0.04% . We also added: “(INPs were not included in this calculation, accounting for which would result in an even lower frequency).”

4) Figure 2A should have a key for the colored boxes: what do each represent?Also, in Figure 2 there is small number of green cells that can be seen in both FOFO1 and FOFO2 non heat shocked samples. These do not seem to cause a problem since tumors only form when pros or brat are removed in progenitors. For other applications this leakiness (which most likely results from heat shock promoters leakiness) can be problematic. This should be discussed and quantified.

Colored boxes in Figure 2A were the same as had been detailed in Figure 1A; we have now indicated that in Figure 2A legend.

Regarding single EGFP-positive cells in the absence of heat shock in Figure 2B, we considered it might be an analogous phenomenon to that frequently observed in UAS-reporter lines in the absence of GAL4 – suggesting reporter expression independent of the intended promoter. We have quantified these as appearing at a frequency of 0.25-0.33 cells per brain lobe across all 6 genotypes with no significant difference between them (n=240 brain lobes total) and have added the following: “Occasional single cells labeled with EGFP could be seen in the absence of hs (average of 0.3 per brain lobe; n=240 pooling data for FOFO1.0 and FOFO2.0 carrying *CD2^miRs^, pros^miRs^* or *brat^miRs^* with no significant difference between genotypes).”

*5) The authors aim to obtain reproducibility of tumor induction but never really compare the tumors in different animals. Is the increase in number of NSC similar between samples induced the same way? How does it change when heat shock regime is changed? These should be quantified.*

The number of NSCs in tumors is too large to count so we measured tumor volume. We performed measurements of EGFP-positive tumor volumes obtained with single and double 1.5 h heat-shock regimes on the *enhancer-FLP; hsmFLP5, FOFO brat^miRs^* stock crossed out to the control strain *w^1118^*(to mimic having a single copy of each recombinase and of FOFO as would be the case in a screen) as well as with the double 1.5 h heat-shock on the compound stock homozygous for *enhancer-FLP* (on X) thus carrying 2 copies of FLP as another way of loading more FLP into cells. We found a significant increase in mean tumor volume with an increase in recombinase loading. Mean tumor volume does vary between animals even when both recombinases are loaded “twice”, presumably due to exponential growth of tumor cells. However, their distribution follows a normal distribution in double heat shocked single and double FLP copy animals but not in the single heat shocked single FLP copy animals In a genetic screen, we would judge the candidate modifier on its ability to shift the distribution of mean tumor volume. We also assessed tumor reproducibility as the variation in tumor size between brain lobes (in proportion to the sum volume of tumor burden in both lobes per animal since the absolute values varied greatly), calculated for each animal for each condition. Normalized variation between tumor volume decreased significantly with the increase in recombinase loading, i.e., between single and double heat shock regimes (1 versus 2 loads of mFLP5) in animals with a single FLP copy; and between animals subject to the same heat shock regime when two genomic copies of FLP were provided instead of one. We added representative images of specimen generated with each of the regimes above, and quantifications to Figure 6. In light of these results, we recommend generating stocks with two copies of the recombinases when reproducibility of phenotype domain is key; and have edited/added the following to the Results: “Concerning reproducibility, in the first instance we were looking for symmetry between brain lobes, suggestive of near-complete extent of recombination within the enhancer domain. […] The double-load of mFLP5 and of FLP(D) greatly reduced tumor asymmetry between lobes (Figure 6C).”

We also added to the Discussion: “The degree of tumor reproducibility reported differences in expression dynamics of the lineage-restricted enhancers (e.g., as seen by more asymmetric *brat* tumors with *R19H09-FLP(D)* than with *stg^14^FLP(D)*) and incomplete extent of recombination within the enhancer domain. Reproducibility could be improved by increased loading of recombinases so in cases where reproducibility is desired we recommend using multiple copies of recombinase transgenes.”

We also added to the Quantifications and statistics section of Materials and methods: “For Figure 6, a complete Z-stack was acquired for every brain (both lobes). […] Significance of difference between each condition was tested by Kruskal-Wallis with Dunn’s multiple comparisons test as post hoc analysis.”

6) The temporal control of expression is inherently dependent on the expression dynamics of enhancers. Judging from Figure 4—figure supplement 1, almost all the enhancers that the authors use unlock the construct in a larger set of cells that end up expressing the enhancer at third instar (comparing enhancer-GAL4 Tub>Stop>GAL4, UAS-FLP mediated GFP expression to direct enhancer-GAL4 mediated expression). Therefore both the temporal control and special control is limited. The wording in the text should be toned down when it comes to spatial and temporal control of expression and caveats should be more clearly stated.

Indeed, it is precisely because no snapshot of direct enhancer-GAL4 expression can be relied upon to report the complete expression domain (due to dynamic variations over time) that we added images of FLP-out for comparison with images of “snapshot” expression at third instar expression (Figure 4—figure supplement 1). We have now added the following: “Spatiotemporal control is constrained by the dynamics of the enhancer in *enhancer-FLP*.”

7) The heat shocks are conducted so early for tumorigenesis and conducting two heat shocks completely ablates the temporal control aspect in the experiments. Is this tumor induction method better/more robust than a single flip out regulated by enhancer specific Flp? A head to head comparison should be included.

A screening phenotype can be induced with FOFO with a single heat-shock and for many applications this will be fine as reproducibility of screening phenotype volume is not necessary (e.g., studying dendritic branching pattern). When studying growth, the latter is desirable and depends on the extent of recombination within the enhancer-FLP domain. This in turn depends on recombination efficacy (we said: “The degree of reproducibility of FOFO-induced tumors depends on reproducibility of the expression domain of FLP, the strength of this expression and recombination efficiency”). The double heat shock regime (providing two loads of hs-mFLP5) led to increased tumor volume reproducibility relative to a single heat shock, as did providing two copies of FLP. As stated in response to point 5 above, we have now added this data to Figure 6 and reflected upon it in the Discussion. In our case, the timings of the double heat shock regime were carefully planned to *not* ablate temporal control over NSC tumor induction as both were performed during the time-window when NSCs are quiescent and thus with no intervening divisions. Therefore, what was achieved was merely two opportunities for loading cells with mFLP5 and only after both would tumor induction initiate (following NSC reactivation from quiescence).

*8) The authors should deposit their plasmids in Addgene or BDGRC and explain how to expand applications for different applications by customizing their tools. They should also deposit their transgenic flies in the BDSC. They nowhere mention that they are planning or willing to do this. These reagents and stocks should be deposited before the paper is accepted or at the time the paper is going to press.*

We will deposit plasmids and transgenic flies in stock centres upon paper acceptance and have added the following at the end of the Materials and methods section: “Reagent availability. Plasmids and transgenic flies are deposited in stock centres.” We are planning to deposit the following (please let us know if there are others you would like):

Plasmids:

. pFOFO2.0-CD2^miRs^-GAL4^miRs^

. pDEST-Hemmar-FLP(D). pDEST-Hemmar-DSCP-ftz-FLP(D)

Transgenic flies:

. *hs-mFLP5* (attP16)

. 9 *enhancer-FLP(D)* (attP18) generated with pDEST-Hemmar-FLP(D)

. *FOFO2.0-CD2^miRs^-GAL4^miRs^* (attP40)

. *FOFO2.0-pros^miRs^-GAL4^miRs^* (attP40)

. *FOFO2.0-brat^miRs^-GAL4^miRs^* (attP40)

Regarding expanding the tool-kit for different applications, we have now added: “Control flies (those with miRs against CD2) will be available ‘off the shelf’ and experimental ones can be generated by either gene synthesis or modification of the control plasmid; or by CRISPR-modification of control host flies. […] Other recombinase pairs could be employed where mFLP5/FLP cross-reactivity is a concern.” The sentence in the Discussion, was changed from “Within the CNS, other examples include […]” to “Within the CNS, other applications include […]”.

[Editors' note: further revisions were requested prior to acceptance, as described below.]

The manuscript has been improved but there are some remaining issues that need to be addressed before acceptance, as outlined below:One of the reviewers wrote the following:The Abstract and end of the Introduction still makes it look like this method can be used "off the shelf" when in fact any different application from their example requires genome engineering to replace their example microRNAs with different microRNAs or misexpression ORFs. So I would like to see the Abstract changed from "Altogether, our design opens up efficient genome-wide screens on any deleterious phenotype." to "Altogether, our design opens up efficient genome-wide screens on any deleterious phenotype, once genome engineering is used to place the desired miRNA or ORF into our genotype.We agree that this change should be implemented.Also, we want to ensure that all the reagents will be available from the BDSC without MTA at publication time.

Thank you for your constructive feedback to improve our manuscript and that also provided an opportunity for us to clarify some points further. We have now added in the Abstract the following:

“Altogether, our design opens up efficient genome-wide screens on any deleterious phenotype,once plasmid or genome engineering is used to place the desired miRNA(s) or ORF(s) into our genotype.”

Fly stocks have now been posted to the BDSC and plasmid details have been uploaded to Addgene (we are now awaiting their package for plasmid collection).